# Pediatric Urachal Anomalies: Monocentric Experience and Mini-Review of Literature

**DOI:** 10.3390/children9010072

**Published:** 2022-01-05

**Authors:** Matthias Nissen, Phillip Rogge, Volker Sander, Mohamad Alrefai, Anna Romanova, Ralf-Bodo Tröbs

**Affiliations:** 1Department of Pediatric Surgery, Marien Hospital Witten, St. Elisabeth Group, Ruhr-University of Bochum, Marienplatz 2, D-58452 Witten, Germany; phillip.rogge@elisabethgruppe.de (P.R.); volker.sander@elisabethgruppe.de (V.S.); mohamad.alrefai@elisabethgruppe.de (M.A.); Anna.romanova@elisabethgruppe.de (A.R.); 2Department of Pediatric Surgery, St. Johannes Hospital, Helios Group, An der Abtei 7-11, D-47166 Duisburg, Germany; troebs@icloud.com

**Keywords:** urachal anomalies, pediatric, complication, abscess formation, peritonitis, conservative treatment

## Abstract

Background: Surgery is the current mainstay for the treatment of urachal anomalies (UA). Recent literature data support the theory of a spontaneous resolution within the first year of life. The aim of this study, comprising solely surgically treated children, was to identify age specific patterns regarding symptoms and outcomes that may support the non-surgical treatment of UA. Methods: Retrospective review on the clinico-laboratory characteristics of 52 children aged < 17 years undergoing resection of symptomatic UA at our pediatric surgical unit during 2006–2017. Data was dichotomized into age > 1 (*n* = 17) versus < 1 year (*n* = 35), and complicated (pre-/post-surgical abscess formation or peritonitis, *n* = 10) versus non-complicated course (*n* = 42). Results: Children aged < 1 year comprised majority (67%) of cohort and had lower complication rates (*p* = 0.062). Complicated course at surgery exclusively occurred in patients aged > 1 year (*p* = 0.003). Additionally, complicated group was older (*p* = 0.018), displayed leukocytosis (*p* < 0.001) and higher frequencies regarding presence of abdominal pain (*p* = 0.008) and abdominal mass (*p* = 0.034) on admission. Regression analysis identified present abdominal pain (OR (95% CI), 11.121 (1.152–107.337); *p* = 0.037) and leukocytosis (1.435 (1.070–1.925); *p* = 0.016) being associated with complicated course. Conclusions: This study provides evidence that symptomatic disease course follows an age-dependent complication pattern with lower complication rates at age < 1 year. Larger, studies have to clarify, if waiting for spontaneous urachal obliteration during the first year of life comprises a reasonable alternative to surgery.

## 1. Introduction

The estimated prevalence of urachal anomalies (UA) in the pediatric population is 1.03 to 1.6%. The urachus represents an embryological remnant of the cloaca and the allantois. This tubular connection between the primitive bladder dome and the umbilicus normally involutes around the fourth or fifth gestational month, forming the median umbilical ligament in the preperitoneal space of Retzius [1]. Obliteration of the lumen may be disturbed at any point along the tract, resulting in different morphological variants of urachal anomalies [2]. Currently, there is no clear consensus regarding the optimal management of pediatric UA, especially in asymptomatic cases. Surgical removal has been advocated as a standard of care for years, preventing its potential malignant transformation later in life or its symptomatic recurrence [1]. More recently, others proposed a delayed surgery after failed resolution of patent urachus on repeated imaging after six to twelve months of age, or surgical intervention in symptomatic cases only [3]. However, even if infection occurs, surgical therapy not always seems to be mandatory, since a subset of infected UA cases have been identified as being susceptible to antibiotic regime alone [4,5,6]. The purpose of this study was to assess the age-stratified differences in clinico-laboratory factors defining pre- and postoperative complications in a solely surgically treated cohort. In this context, identification of factors associated with complicated UA, defined as pre- or postoperative abscess formation or peritonitis, might add knowledge regarding the optimal treatment of children with UA.

## 2. Methodology

### 2.1. Patients

In this study, 64 consecutive cases, aged one day to 16 years with the diagnosis of UA were identified according to the International Classification of Diseases system (10th Revision; ICD-10 code Q64.4). Exclusion criteria were any co-associated malformations other than UA alone, or its incidental detection and removal under concurrent surgery for another reason as follows: ileal atresia with perforation (*n* = 1), Meckel’s diverticulum (*n* = 1), omphalomesenteric duct resection with bowel anastomosis (*n* = 1), laparoscopic inguinal herniorrhaphy (*n* = 3), infantile hypertrophic pyloric stenosis (*n* = 2), open inguinal and umbilical hernia closure (*n* = 2), anorectal reconstruction procedure at combined anal atresia and Müllerian aplasia (*n* = 1), and appendicitis (*n* = 1). Surgeries were performed at our Pediatric Surgery Department during the study period January 2006—January 2017.

### 2.2. Study Design

In this retrospective, monocentric study, the estimated parameters included the patients’ biometric and laboratory data, the histological workup of the urachal specimen and procedural parameters. The primary outcome measure of this study was the detection of age-differences within the complicative and non-complicative group. Secondary outcome measures included identification of age-stratified differences regarding clinico-laboratory and procedural parameters, as well as the assessment of the pre- and postoperative complication spectrum.

### 2.3. Surgery

UA was confirmed by histopathology and/or by surgery in all cases. No structural urinary tract malformation, nor recurrent urinary tract infections were identified by medical history. Irrespective of its inflammatory state, in each case (*n* = 52), a single-staged open surgical approach with primary UA excision was performed by an infra-umbilical, semicircular incision. The UA was separated from the umbilical border and, if required, further exposed by a T-shaped division of the linea alba from the umbilical level along towards the urachal insertion at the bladder dome. Depending on the subtype, en bloc excision of UA with or without bladder cuff was performed. Complicative disease course (*n* = 10) was defined as any occurrence of pre- or post-surgical abscess formation, or peritonitis related to UA. The postsurgical complications were graded according to the Clavien-Dindo system, as originally described in 2004 [7]. Of the five cases with postsurgical complications (Table 1), three required surgical intervention under general anesthesia (Grade IIIb) and two were treated conservatively by either opening of the infected wound at the bedside alone at day eleven after the first surgery (Grade I), or by additional administration of antibiotics (Grade II) at day nine after the first surgery. The cases graded IIIb requiring redo surgery for abscess formation comprised two cases due to incomplete urachal resection 238 and 449 days after the first surgery, and one case with surgical site infection 20 days after the initial surgery. 

### 2.4. Statistical Analysis

Data sampling and analysis were performed with Microsoft Excel© (version 2010, Microsoft Corporation, Redmond, WA, USA), SPSS^®^ (version 27, SPSS Inc., Chicago, IL, USA) and OriginPro© (version 2021, OriginLab, Northampton, MA, USA). Normal distribution was confirmed by Kolmogorov-Smirnov testing. Categorical variables were presented as frequencies (percentages) and compared by Fisher’s exact testing. Comparison of paired nonparametric variables, expressed as medians with 1st and 3rd quartiles (Q1–Q3), was performed by Mann-Whitney U test. Normally distributed variables were given as means ± standard deviation (SD) and compared by Student’s t-test. Areas under the curve (AUC) of receiver operating characteristic (ROC) curves were utilized and the Youden index [8] was applied. Sensitivity, specificity and odds ratio (OR) were calculated for chosen cut-off values. As described elsewhere [9], variables with AUC > 0.7 were considered eligible for regression analysis. A *p*-value ≤ 0.05 was considered statistically significant. 

## 3. Results

### 3.1. Clinical Findings

Basic data of investigated parameters are presented in Table 1. A total of 52 patients with surgical removed UA were included with a slight male preponderance rate of 1.2:1. On histopathological differentiation, patent urachus (*n* = 23; 44%), umbilical-urachal sinus (*n* = 21; 41%) and urachal cyst (*n* = 8; 15%) were identified. Umbilical discharge was the leading symptom in 44/52 (85%), followed by umbilical erythema in 31/52 (60%). 

### 3.2. Age-Dependent Characteristics

Regarding age-dependent differences (Table 1), patients below one year comprised the majority of cases (35/52; 67%) with evenly distributed gender, whereas male preponderance (65%) was observed in the age group above one year. Regarding symptoms, a predominance of abdominal pain on admission in age groups above one year (*p* < 0.001) was obtained with no further differences. The total rate of patients with complications was 10/52 (20%), with each 5/52 (10%) being observed either prior to or after surgery. Pre-surgical complications were exclusively obtained in age class above one year (*p* = 0.003) with rates of 3/5 (60%) regarding peritonitis and 2/5 (40%) regarding pre-fascial abscess formation. In addition, peritonitis at surgery exclusively occurred in patients aged above one year (*p* = 0.031). Post-surgical complications occurred in 4/5 (80%) of patients aged below one year and in 1/5 (20%) of patients aged above one year. The post-surgical overall infection rate was 1/17 (6%) at ages above and 4/35 (12%) at ages below one year. The healthcare insurance status of the patients’ guardians, as an indirect measure of the socioeconomic status, did not differ between those who presented their children prior to the age of one year (privately insured: 3/32 (9%)) versus those who presented their children beyond the first year of life (privately insured: 2/15 (12%)).

### 3.3. Complications

Compared with non-complicated control (Table 2), complicated group was older (*p* = 0.018) with higher body weight (*p* = 0.013) and higher rates of both present abdominal pain (5/10 vs. 4/42; *p* = 0.008), and abdominal mass (2/10 vs. 0/42; *p* = 0.034) on admission. Leukocytosis on admission was also observed in complication group (*p* < 0.001), being paralleled by a trend towards raised CRP (*p* = 0.20). While surgery duration did not differ (*p* = 0.34), post-surgical hospitalization was prolonged in complicated cases (*p* = 0.016). 

Applied ROC curve analysis for differentiation of complicated from non-complicated group (Table 3 and Figure 1) identified leukocytosis ≥ 14.3 × 10^9^/L (*p* = 0.002), age ≥ 5 months (*p* = 0.017) and weight ≥ 6.2 kg (*p* = 0.012) on admission as predictors for complicated UA course. Due to the high level of multicollinearity regarding age and body weight, the latter was excluded from the final model that included age, leukocyte levels, presence of abdominal pain, and abdominal mass, respectively. On forward (Likelihood-Ratio) binary logistic regression analysis, presence of abdominal pain (OR (95% CI), 11.121 (1.152–107.337); *p* = 0.037) and leukocytosis (1.435 (1.070–1.925); *p* = 0.016) on admission were independent predictors for a complicated disease course.

## 4. Discussion

Children aged below one year comprised the majority of enrolled patients with a trend towards lower rates of overall complications in this age class. Pre-surgical complications in symptomatic UA exclusively occurred in age class above the first year of life. There was a trend towards more frequent post-surgical complications within the first year of life (Table 1). Moreover, we identified a subset of factors on admission that were characteristically for the complicated course of UA, namely a higher rate of present abdominal pain and abdominal mass, together with an altered inflammatory state, as demonstrated by elevated levels of WBC and (in tendency) by an elevated CRP. Moreover, leukocytosis and the presence of abdominal pain were independently associated with complicated UA. In addition, obtained older age in complicated compared to non-complicated UA was in support of findings on a lowered overall (in tendency) and pre-surgical occurrence of complicated UA in the age group below one year of life. 

The current trend towards the change of management of UA was based on findings that UA seemed to be more frequent than historically thought, possibly due to improvements in ultrasound (US) diagnostics, that made detection of UA more likely. In this context, based on US reviews, Robert et al. reported that UA was present in the majority of patients below 16 years with a decrease in frequency with increasing age, supporting the theory of a physiological regression of UA with age [10]. In several reports, spontaneous and perhaps postponed physiological involution of UA had been increasingly observed [6]. In a review of 3400 US reports, Ueno et al. identified 56 cases (1.7%) with UA. Of 44 cases treated conservatively, spontaneous resolution of UA was documented in nine cases within a follow-up period of 12 ± 10 months [5]. In a retrospective study on 23 patients, Galati et al. found a spontaneous resolution in 50% of UA with emphasis on ages below six months, in which 80% resolved conservatively [6]. Of 15 patients with symptomatic UA, as retrospectively reviewed by Lipskar et al., seven were managed conservatively with no recurrences on 26 months follow-up [11]. Accordingly, in a retrospective study on repeated imaging in 103 patients with UA, Naidich et al. observed a spontaneous resolution of UA in 15 patients out of 19 (79%, symptomatic *n* = 8, non-symptomatic *n* = 7) [4]. Finally, and also in line with previous reports, Stopak et al. reported on a spontaneous resolution in 87% (*n* = 13) of patients with conservatively monitored UA within about one year of diagnosis [12]. Moreover, patients aged below six months comprised 92% (*n* = 12) of cases in which UA resolved with observation. Given these literature data, a delayed obliteration at age of six to twelve months of life might be considered physiological and surgical intervention should be reserved only for cases of multiple symptomatic episodes, visible umbilical urine drainage, peritonitis, or abscess formation [12]. Given our observed age-pattern with a pre-surgical complicated course exclusively observed in children older than one year and a trend towards elevated post-surgical complication rates in patients younger than one year, a conservative approach in children during their first year of life might be considered in line with a proposed conservative approach. However, if a child younger than one year was treated conservatively, follow up at age of one year and elective surgery would be mandatory if UA was persisting. Noteworthy, the hypothesis of spontaneous urachal involution has not yet been verified by larger, histologically confirmed studies.

The following general considerations regarding surgery and anesthesia are inherent to and in support of a proposed conservative approach in this vulnerable age group. In general, a temporarily diminished neurodevelopmental outcome following surgery in children cannot be excluded, as described by Walker et al. on reduced fine and gross motor subscales in 24 children at one year of age who underwent pyloromyotomy [13]. Moreover, in a prospective study by the same group, infants with pyloric stenosis showed lowered cognitive, receptive language, and motor subscales secondary to surgery when compared with a healthy control group at one year of age [14]. In addition, DiMaggio et al. described that children undergoing inguinal hernia repair during the first three years of life were more than twice as likely diagnosed with a developmental or behavioral disorder compared to a reference group. These results raised concern towards an association of early surgery and exposure to potentially neurotoxic anesthetic agents [15]. In this context, animal studies strongly suggested that neurodegeneration with possible long-term cognitive sequelae might occur secondary to exposure with higher concentrations of anesthetic agents. However, their in-vivo influence, especially on neonates and infants, remains unknown [16]. With these observations in mind, any surgical indication, especially in the youngest patients, should be considered on an appropriate and risk-adapted basis.

Knowledge on the negligible role of retained urachal anomalies in developing future urachal carcinogenesis [1,17] may also be considered in agreement with a conservative approach of UA during the first year of age. 

Contrasting to other studies with lower post-surgical complication rates of 7.7%, as published by McCollum et al. [18], and 6.7% in the study by Cilento et al. [19], Naiditch et al. recorded a complication rate of 14.7% [4]. This disparity might partly be based on a prolonged mean follow up period of 26 months, while follow up period was not documented in both former studies. However, our post-surgical complication (wound infection) rate of 10% lies in between these values (Table 1). In accordance with Naiditch et al., our finding of delayed onset of post-surgical complications with a mean of 145 ± 196 days (median 20 days; min. 9 days; max. 449 days) supported their claim towards a prolonged follow-up period after resection of UA. Noteworthy, as also applied by Naiditch et al. [4], our single-staged surgical approach itself may have affected outcome towards higher postoperative complication rates, since authors as McCollum et al. [18], or Minevich et al. [20] have reported on lower complication rates if definite (elective) excision was preceded by primary abscess drainage (two-staged procedure).

It remains speculative to what extent a lowered level of disease awareness, as sometimes being ascribed to a lowered socioeconomic state may have contributed to a delayed initial presentation in the age group above one year. This is due to first, the retrospective study design and second, to the fact that our hospital admission forms did not reliably determine the patients’ socioeconomic status. However, according to some authors [21], an individual’s socioeconomic state might (to some extent) be reflected by its health insurance state. Since no differences in health insurance status (private versus statutory) were discerned within age groups, the theory of an assumed lowered socioeconomic state in those patients with delayed presentation cannot be maintained by our data. 

### Limitations

This study had several limitations. The main limitation was the lack of a conservatively treated control group. Another one was the single-institutional, retrospective design with relatively small sample size, thus being subject to various unaccounted confounders. Though being of diagnostic importance, documentation of obtained urinalysis and microbiological workup was rudimentary, and therefore, not included in this study.

## 5. Conclusions

In conclusion, our results on a surgically treated cohort with symptomatic UA illustrated an age dependent complication pattern with, inter alia, pre-surgical peritonitis occurring exclusively in patients older than one year, and a trend towards more postoperative complications in patients below one year. Due to the study design, no definite conclusions could be drawn from our data. Therefore, it remains subject to larger, multicentric studies to clarify if waiting for spontaneous urachal obliteration during the first year of life constitutes an alternative to surgery.

## Figures and Tables

**Figure 1 children-09-00072-f001:**
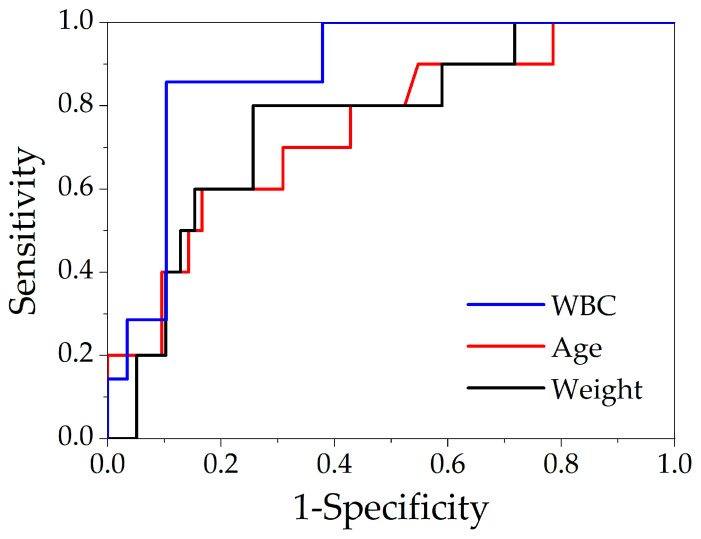
ROC-curves for differentiation of complicated from non-complicated disease course in urachal anomalies (WBC White blood cell count).

**Table 1 children-09-00072-t001:** Age dependent characteristics of urachal anomalies.

	Σ	Age < 1 Year	Age > 1 Year	*p*
*n*, (%)	52	35 (67)	17 (33)	
Age (months)	3 (2–33)	2 (1- 3)	134 (33–148)	<0.001 ^‡^
Body weight (kg)	13.4 ± 16.8	5.2 ± 1.3	34.0 ± 20.0	<0.001 ^†^
Body length (cm)	84 ± 41	57 ± 6	134 ± 32	<0.001 ^†^
Body mass index (kg/m^2^)	16.3 ± 2.8	15.5 ± 1.9	17.9 ± 3.5	0.037 ^†^
Gender (f:m)	24:28	18:17	6:11	0.38 *
Body temperature on admission (°C)	37.0 ± 0.5	37.0 ± 0.4	37.1 ± 0.7	0.78 ^†^
Health insurance state *n*, (%)				
private/statutory	5/47	3/32 (9)	2/15 (12)	>0.99 *
Symptoms on admission *n*, (%)				
-umbilical discharge	44 (85)	31 (89)	13 (77)	0.41 *
-abdominal pain	9 (17)	0	9 (53)	0.001 *
-abdominal mass	2 (4)	0	2 (12)	0.10 *
-erythema	31 (60)	20 (57)	11 (42)	0.77 *
-fever	2 (4)	0	2 (12)	0.10 *
-dysuria	1 (2)	0	1 (6)	0.33 *
Procedural				
-post-surgical hospital stay	3 (2–4)	2 (2–4)	3 (2–4)	0.91 ^‡^
-surgery duration	63 ± 38	58 ± 36	72 ± 42	0.24 ^†^
Overall complications *n*, (%)				
Σ	10 (20)	4 (12)	6 (36)	0.062 *
-pre-surgical onset *n*, (%)	5 (10)	0	5 (30)	0.003 *
peritonitis	3 (6)	0	3 (18)	0.031 *
pre-fascial abscess	2 (4)	0	2 (12)	0.10 *
-post-surgical onset *n*, (%)	5 (10)	4 (12)	1 (6)	0.66 *
peritonitis	1 (2)	1 (3)	0	>0.99 *
pre-fascial abscess	4 (8)	3 (9)	1 (6)	>0.99 *
Recurrence management *n*, (%)				
Σ	5 (10)	4 (12)	1 (6)	0.66 *
-conservative	2 (4)	2 (6)	0	0.55 *
-surgical	3 (6)	2 (6)	1 (6)	>0.99 *

Variables were compared by Fisher’s exact test (*), by Mann-Whitney U test (^‡^), or by Student’s *t* test (^†^); *p* ≤ 0.05 was defined as significant.

**Table 2 children-09-00072-t002:** Pre- and postoperative complicated versus non-complicated course of urachal anomalies.

	Complicated	Non-Complicated	*p*
*n*, (%)	10 (19)	42 (81)	
Age (months)	42 (2–147)	2 (1–17)	0.018 ^‡^
Body weight (kg)	14.9 (6.0–39.5)	5.7 (4.5–8.2)	0.013 ^‡^
Body length (cm)	122 (61–158)	61 (54–90)	0.07 ^‡^
Body mass index (kg/m^2^)	16.9 ± 0.8	16.2 ± 3.1	0.24 ^†^
Gender (f:m)	6:4	18:24	0.48 *
Body temperature on admission (°C)	37.2 ± 0.8	37.0 ± 0.5	0.60 ^†^
Symptoms on admission *n*, (%)			
-umbilical discharge	7 (70)	37 (88)	0.17 *
-abdominal pain	5 (50)	4 (10)	0.008 *
-abdominal mass	2 (20)	0	0.034 *
-erythema	6 (60)	23 (55)	>0.99 *
-dysuria	1 (10)	0	0.19 *
Laboratory on admission			
CRP (mg/dL)	5.8 ± 9.7	0.4 ± 0.8	0.20 ^†^
WBC (x10^9^/L)	16.7 ± 3.5	10.8 ± 3.7	<0.001 ^†^
Platelets (x10^9^/L)	483 ± 151	459 ± 209	0.79 ^†^
Hematocrit (%)	36 ± 5	37 ± 13	0.77 ^†^
Hemoglobin (g/dL)	12.3 ± 1.6	13.2 ± 2.9	0.45 ^†^
Procedural			
Post-surgical hospital stay	4 (3–9)	2 (2–4)	0.016 ^‡^
Surgery duration	73 ± 50	60 ± 35	0.34 ^†^
Diagnostic mode *n*, (%)			
Ultrasound	6 (60)	25 (60)	>0.99 *
Surgery	4 (40)	17 (40)	>0.99 *

CRP C-reactive protein; WBC White blood cell count. Variables were compared by Fisher’s exact test (*), by Mann-Whitney U test (^‡^), or by Student’s *t*-test (^†^); *p* ≤ 0.05 was defined as significant.

**Table 3 children-09-00072-t003:** ROC-curve data of selected variables and their cut-off values indicative of complicated disease course regarding urachal anomalies.

	AUC (**±**SE)	Cut-Off	Sensitivity	Specificity	95% CI	*p*	OR (95%CI)
WBC	0.88 ± 0.09	≥14.3 × 10^9^/L	86%	90%	0.71–1.06	0.002	52 (4.57–591.30)
Age	0.74 ± 0.09	≥5 months	70%	69%	0.58–0.91	0.017	5.2 1(1.16–23.39)
Weight	0.76 ± 0.09	≥6.2 kg	80%	74%	0.58–0.94	0.012	11.6 (2.10–64.02)

AUC Area under the curve, SE Standard error; OR Odds ratio, CI Confidence interval; WBC White blood cell count; *p* ≤ 0.05 was defined as significant.

## Data Availability

The raw data supporting the conclusions of this article will be made available by the corresponding author M.N., without undue reservation.

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
