# Peer review of "Pediatric Urachal Anomalies: Monocentric Experience and Mini-Review of Literature"

_children, 2022, doi:10.3390/children9010072_

Round 1

Reviewer 1 Report

The authors did a monocentric retrospective review study named: "Pediatric urachal anomalies: Monocentric experience and 2 mini-review of literature"

Few of my remarks:

INTRODUCTION:

- Very nicely written.

MATERIALS AND METHODS:

- There is a sentence: "Surgery was performed between January 2006 and January 2017 at our Pediatric Surgery Department.". Please rewrite the sentence into: "study period was..." or "surgeries / operations were performed..."

-  Please add and clearly state primary and secondary outcomes.

RESULTS:
- I would recommend the data table of input data (demographic, etc...) be relocated into materials and methods.

- Table 1 is separated! The title and the first row are on the previous page, please make a space between them so the table is more reader-friendly.

- Same table 1, in the bottom please rewrite the legend! The bottom of the table should state which test/s was/were used for every calculated or numerated variable and put with different signs (*, or letters in superscript). In all other tables should be done the same, the long text should be avoided and changed into short descriptions of text and legends.

- I suggest the description of how the specific complication was treated and stratified into Clavien Dindo Classification.

A bit of English proofreading is needed, as it has flawed tenses in text.

All in all, this is a well-writen study, and after revision will be even better.

Wish to congratulate the authors on their work.

And will use the moment to give my best for the Holydays and best of luck, health, and prosperity in the forthcoming year of 2022.

Reviewer 2 Report

The authors clarified the field of surgical treatment of urachal anomalies in a very comprehensive manner. The results are clear and the conclusions are adequate to the results. The discussion is fair and sound. All Tables are presented clearly. References are adequate for the purpose.
